# Separation of Donor and Recipient Microbial Diversity Allows Determination of Taxonomic and Functional Features of Gut Microbiota Restructuring following Fecal Transplantation

Evgenii I. Olekhnovich,[a] Artem B. Ivanov,[b] Vladimir I. Ulyantsev,[b] Elena N. Ilina[a]

[a]Federal Research and Clinical Centre of Physical and Chemical Medicine, Federal Medical and Biological Agency of Russia, Moscow, Russian Federation
[b]ITMO University, Saint Petersburg, Russian Federation

Evgenii I. Olekhnovich and Artem B. Ivanov contributed equally. Vladimir I. Ulyantsev and Elena N. Ilina contributed equally to this work. The order of the authors was determined by the agreement of the parties.

**ABSTRACT** Fecal microbiota transplantation (FMT) is currently used in medicine to treat recurrent clostridial colitis and other intestinal diseases. However, neither the therapeutic mechanism of FMT nor the mechanism that allows the donor bacteria to colonize the intestine of the recipient has yet been clearly described. From a biological point of view, FMT can be considered a useful model for studying the ecology of host-associated microbial communities. FMT experiments can shed light on the relationship features between the host and its gut microbiota. This creates the need for experimentation with approaches to metagenomic data analysis which may be useful for the interpretation of observed biological phenomena. Here, the recipient intestine colonization analysis tool (RECAST) novel computational approach is presented, which is based on the metagenomic read sorting process per their origin in the recipient's post-FMT stool metagenome. Using the RECAST algorithm, taxonomic/functional annotation, and machine learning approaches, the metagenomes from three FMT studies, including healthy volunteers, patients with clostridial colitis, and patients with metabolic syndrome, were analyzed. Using our computational pipeline, the donor-derived and recipient-derived microbes which formed the recipient post-FMT stool metagenomes (successful microbes) were identified. Their presence is well explained by a higher relative abundance in donor/pre-FMT recipient metagenomes or other metagenomes from the human population. In addition, successful microbes are enriched with gene groups potentially related to antibiotic resistance, including antimicrobial peptides. Interestingly, the observed reorganization features are universal and independent of the disease.

**IMPORTANCE** We assumed that the enrichment of successful gut microbes by lantibiotic/antibiotic resistance genes can be related to gut microbiota colonization resistance by third-party microbe phenomena and resistance to bacterium-derived or host-derived antimicrobial substances. According to this assumption, competition between the donor-derived and recipient-derived microbes as well as host immunity may play a key role in the FMT-related colonization and redistribution of recipient gut microbiota structure.

**KEYWORDS** gut microbiota, fecal transplantation, antibiotic resistance, colonization resistance, fecal microbiota transplantation, metagenomics

Address correspondence to Evgenii I. Olekhnovich, jeniaole01@gmail.com.

The gut microbiota is made up of a large community of microorganisms and viruses, which are a key player in the host body metabolism. Metabolic functions of the gut microbial consortia are associated with support of physiological homeostasis, synthesis of vitamins and amino acids, short-chain fatty acids, and other essential functions (1).

Development of gut microbiota may depend on important events, a few of which can be distinguished: the way of birth (vaginal or cesarean section), maternal microbiota transmission, feeding (breastfeeding or artificial) (2–4), and early antibiotic therapy (5). Also, microbes could enter the intestine from the environment with food (6) and drinking water (7). These factors mediate the formation of a fairly stable gut microbial community that may contain both members in common for different people and unique members.

Fecal microbiota transplantation (FMT) is currently used to treat recurrent *Clostridioides difficile* infection (CDI). Several FMT studies are ongoing in a broad spectrum of disorders. However, neither the therapeutic mechanism of the FMT nor the mechanism that allows the donor bacteria to colonize the intestine of the recipient has been discovered. The changes in the intestinal microbiota under FMT that resulted in colonization with donor bacteria have been described in the case of CDI and metabolic syndrome patients (8–11) and healthy volunteers (12). Nevertheless, only the behavior of donor strains has been demonstrated. Additionally, published approaches do not adequately assess the functional signs of colonization. All of this provides a field for experimentation with approaches addressing these issues.

Here, we present a novel technique that allows the study of recipient gut microbiota reshaping due to FMT—recipient intestine colonization analysis tool (RECAST). This approach is based on the separation of the donor's and recipient's metagenomic reads and allows extraction of read categories by origin: those that came from the donor sample, those that stayed in the recipient intestine, and those with unknown origin. Using the RECAST, we studied the behavior of the donor-derived and recipient-derived microbes after the FMT procedure. Also, we determined which gut microbe features can contribute to the FMT-related restructuring process of human gut microbiota.

## RESULTS

**RECAST algorithm testing using simulated metagenomic data sets.** To check that the RECAST algorithm produces correct read categories, we conducted a series of tests on simulated data of increasing complexity. During the first step of simulations, the set of *Escherichia coli* strain genomes with different nucleotide distances were used. The most probable behavioral scenario that could be observed during FMT has been modeled: donor and recipient strains coexist in the recipient intestine (8). Assessment of classification quality metrics is presented in Fig. S1 in the supplemental material. Using the obtained results, two conclusions can be drawn. First, the quality of the classification depends on the genome coverage by metagenomic reads. Second, given a sufficient number of reads, extremely similar strains (up to 1 − Mash distance = 0.9999) can be distinguished, while strains with lower nucleotide dissimilarity cannot be differentiated even theoretically due to sequencing errors.

According to simulation based on artificial metagenomes (Fig. S2), the classification quality depends on the complexity of the simulated data sets. In addition, the classification quality tending to 100% of precision and recall metrics has been achieved for almost all baskets excluding acquired via FMT, survived during FMT, and external. These observations may be related to variation in read coverage of microbial genomes.

In summary, the classification quality of the RECAST approach depends on the number of microbes within the post-FMT recipient intestinal microbiota and their nucleotide similarity, as well as the coverage of microbial genomes by metagenomic reads. In addition, the RECAST approach can classify donor and pre-FMT recipient metagenomes (colonizer/noncolonizer and resistant/suppressed categories) with high quality, while the classification accuracy of the post-FMT recipient sample (acquired via FMT/common/survived during FMT/external categories) may be reduced due to high genome similarity. In other words, the proposed method allows limited separation of post-FMT bacteria to donor-derived and recipient-derived bacteria when similar strains are present in both donor and recipient samples. Thus, for additional quality control of the

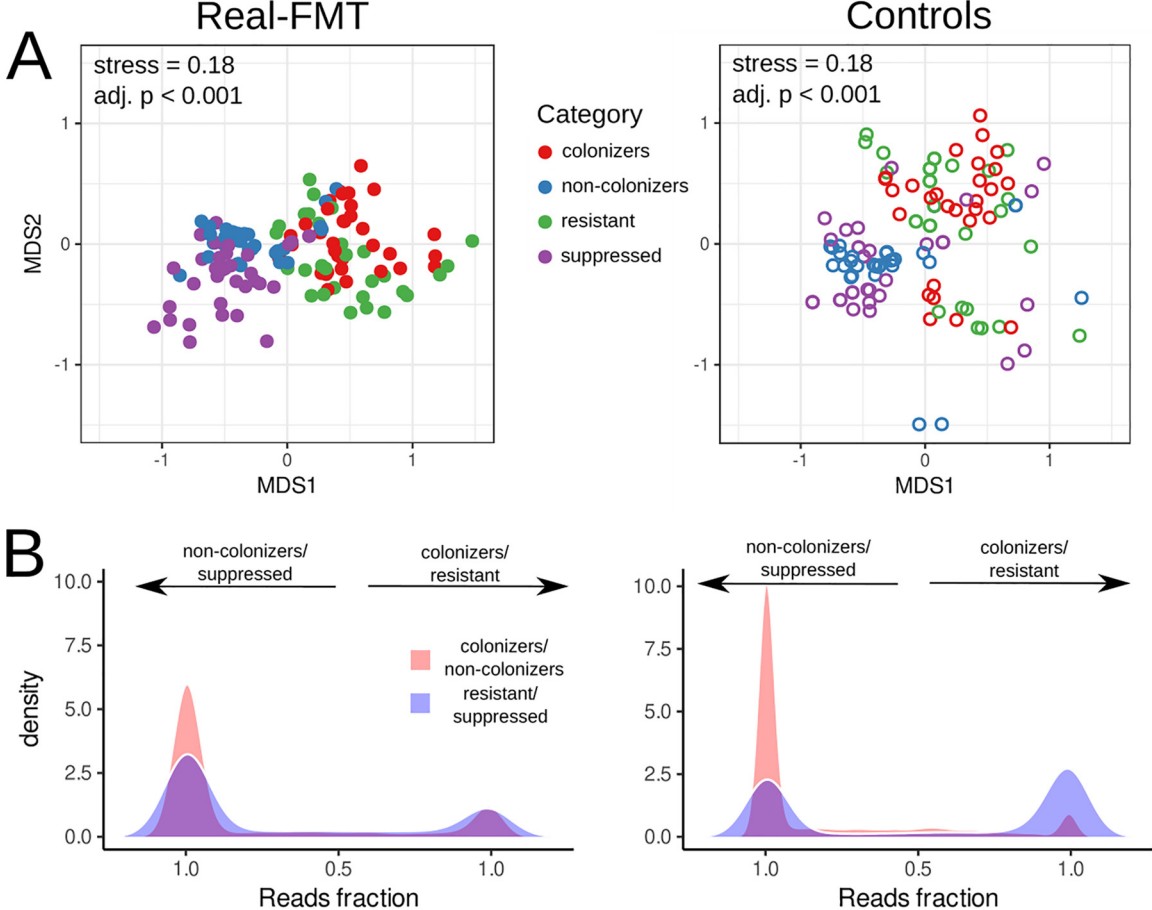

**FIG 1** Taxonomic analysis of the RECAST output obtained using donor and pre-FMT recipient metagenomic samples from both real FMT and control data sets. The left portion of the figure shows results obtained using real FMT data, the right portion shows control data. (A) Nonmetric multidimensional scaling biplots obtained using taxonomic profiles of colonizer/noncolonizer and resistant/suppressed read categories. Bray-Curtis dissimilarity was applied as a comparison measure. The stress values show the difference between distances in the reduced dimension compared to the complete multidimensional space. The adjusted *P* values are shown. (B) Density plots depict microbial read distributions in colonizer/noncolonizer and resistant/suppressed read categories. Extreme points of the *x* axis show the prevailing presence of reads from one bacterium in the corresponding categories. The left peak corresponds to the noncolonizer and suppressed categories, while the right peak corresponds to the colonizer and resistant categories.

classification, a control group including biologically unrelated metagenomes was used in further analysis of real FMT metagenomic data sets. Also, this strategy can be useful to distinguish the effects of FMT and microbiome natural variation.

**RECAST analysis using real metagenomic data sets. (i) Taxonomic analysis of obtained read categories.** To determine the behavior of donor-recipient bacteria after FMT and their distribution in RECAST-produced read categories, taxonomic profiles of read categories were obtained. The colonizer category included donor-derived microbes that were found in the post-FMT recipient's sample. Similarly, the resistant category included recipient-derived microbes that stayed in the recipient's intestine after the FMT procedure. The noncolonizer and suppressed categories included microbes that were not successful in FMT competition.

Nonmetric multidimensional scaling (NMDS) visualization based on taxonomic profiles and Bray-Curtis dissimilarity shows clear separation of the colonizer and resistant categories from the noncolonizer and suppressed categories (Fig. 1A). Additionally, analysis of the variance using permutational multivariate analysis of variance (PERMANOVA) revealed that the read categories were significantly linked to the microbial composition ($R^2 = 0.09$, adjusted $P < 0.001$, Bray-Curtis dissimilarity metric, 10,000 permutations).

Analysis of classified reads separated into different categories also shows a clear difference in microbial composition. The distribution of microbial taxa by read categories

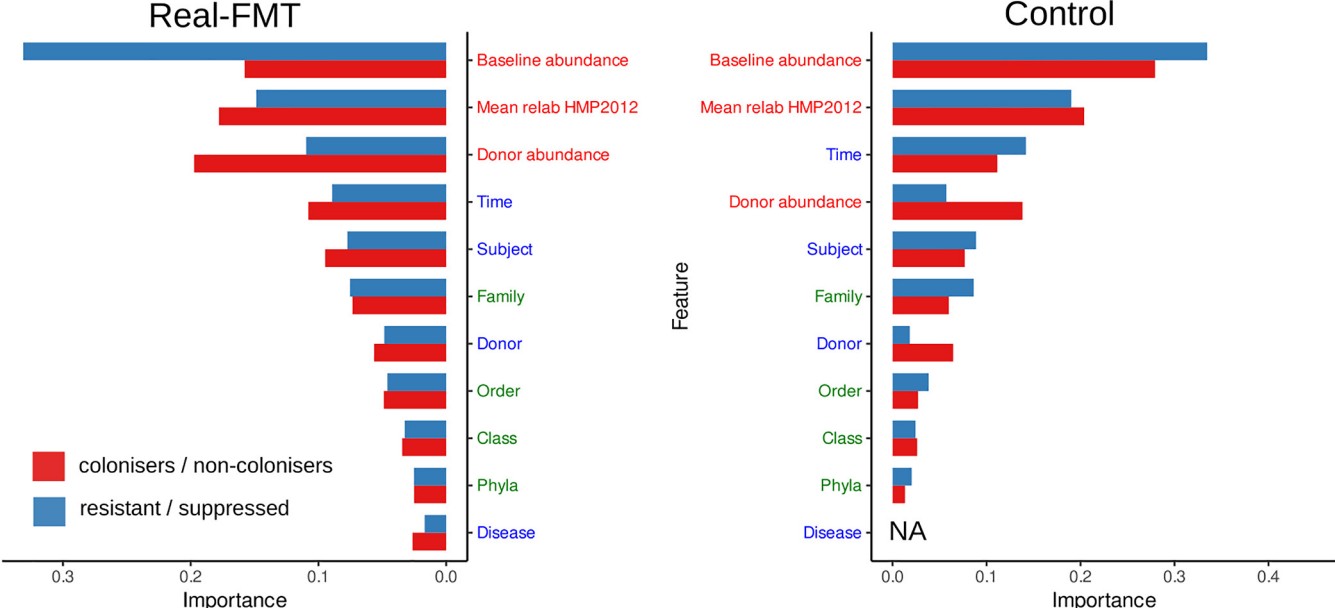

**FIG 2** Random Forest classification importance features of microbes by affiliation with the read category. The classification of the pairs of read categories such as colonizers/noncolonizers and resistant/suppressed was performed separately. The groups of features are shown in different colors. The red color corresponds to abundance-related features, blue color to metadata features, and green color to taxonomy features.

is presented in Tables S3 and S4 in the supplemental material. The uniform separation of reads from one microbe to different paired categories is a minor event (Fig. 1B). There are nearly no reads in the middle of the plot, while reads concentrated on the sides of the plot (for 90% of microbes, the majority [80%] of reads are classified to the one category). Interestingly, the obtained results are similar between real FMT and control data; however, the colonizer categories of control data are substantially smaller than that of real FMT data. This can be explained by the fact that the control data contains biologically independent metagenomic samples. In summary, the RECAST analysis allows the determination of the donor-derived and recipient-derived microbes that can contribute to forming the post-FMT recipient's gut microbiota composition.

**(ii) Discovery of taxonomic and metadata features associated with microbiota restructuring.** To detect specific features related to microbe separation into the different categories, we used the Random Forest algorithm. To perform this analysis, normalized read quantities of microbial species distributed between read categories were used as a predicted variable (this value was used in the analysis presented in Fig. 2B). The taxonomy/metadata features and relative microbial abundances in donor and pre-FMT metagenomic samples were used as predictive features. Additionally, the average relative abundances of microbes from the HMP 2012 data set (Table S5) were added as features in the analysis. Since real FMT metagenomic data were formed from patients' metagenomes with different clinical complications, we added a disease (healthy, *Clostridioides difficile* infection [CDI], or metabolic syndrome [MS]) variable to the analysis.

The classification quality was lower ($0.72 \pm 0.17$ versus $0.89 \pm 0.10$; Wilcoxon rank sum test, $P = 0.02$) in the models based on the real FMT sample set in comparison to the control set (Fig. S3). It may be due to the lack of biological association between control samples. However, the distribution of features for predicting importance was mostly similar for both sets of read categories (Fig. 2).

According to the results of the analysis, the donor-derived bacteria and recipient-derived bacteria that can contribute to the post-FMT recipient metagenomes are associated with higher abundance in the human population gut metagenomes. At the same time, the influence of the donor microbiota for control has been reduced. It is consistent with the lack of biological association between control samples. It is also worth noting the similarity of the results obtained for patients with various diseases (the disease

mSystems®

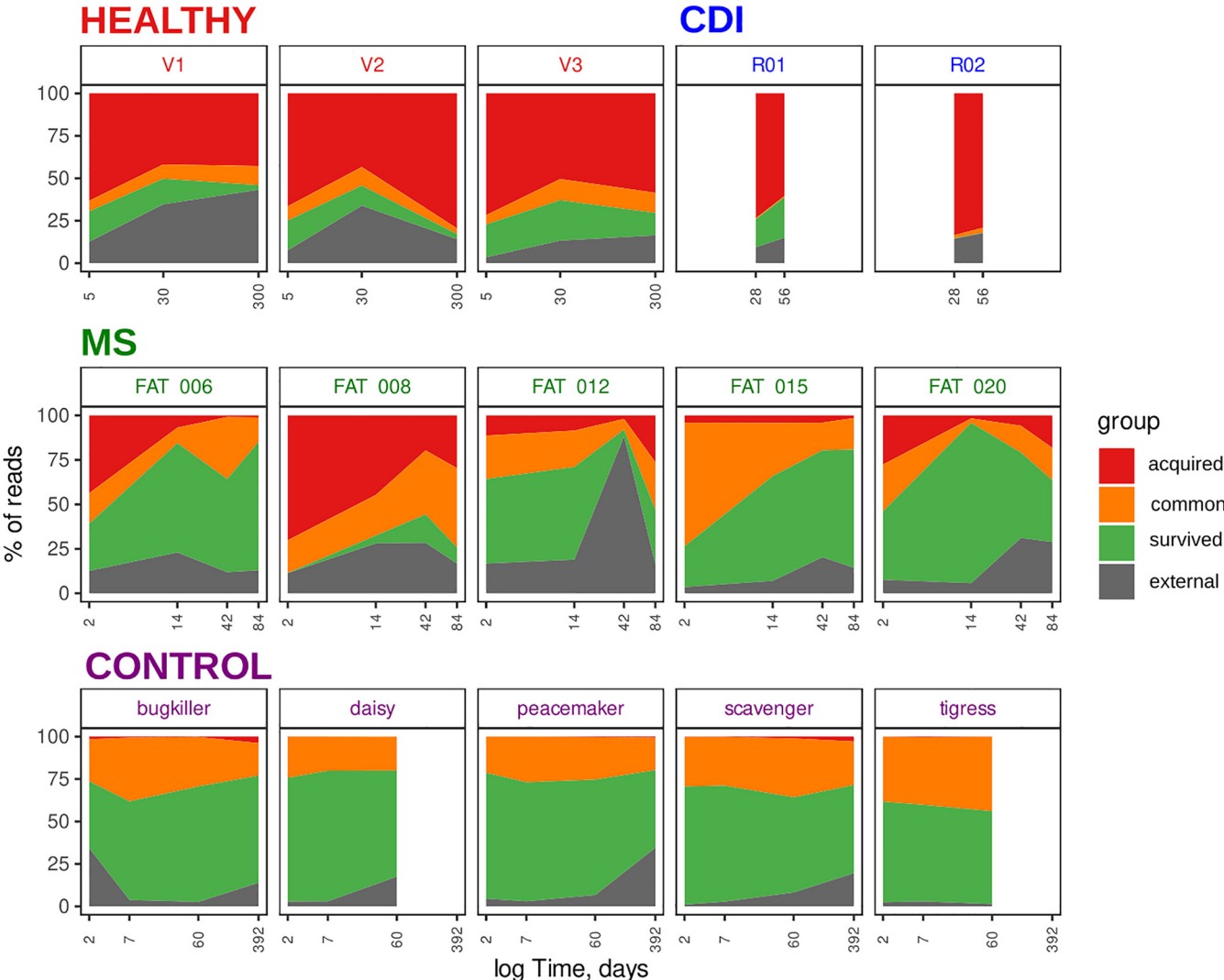

**FIG 3** Area plots show the composition of recipients' post-FMT metagenomic samples over time. the time after elapsed since the FMT procedure. The *y* axis denotes metagenomic read proportion. Only taxonomically annotated reads were counted. The colors show the read category.

variable has the weakest contribution to the prediction). Perhaps, there is a universal feature mediating the human gut microbiota restructuring due to FMT phenomena.

**(iii) Discovery of donor/recipient microbe contribution to recipient microbiome restructuring after FMT.** For determination of donor-derived and recipient-derived microbes' contribution to forming gut microbiota after FMT, donor and pre-FMT recipient samples were queried by post-FMT recipient's metagenomic reads. As a result, new read categories were obtained. These categories included the following categories. The acquired category revealed donor-derived microbes. The common category revealed both donor-derived and recipient-derived microbes. The survived category revealed recipient-derived microbes. The external category revealed microbes (or microbe genome parts) which were not found in the donor and pre-FMT recipient's metagenomes. Also, sorting of the several post-FMT metagenomic samples from the same patient has shown donor-derived and recipient-derived microbial diversity evolution over time. The obtained results are presented in Fig. 3.

The healthy and CDI groups demonstrate dominance of donor-derived microbes in comparison to MS (Wilcoxon rank sum test, $P < 0.001$). At the same time, the recipient's microbiota is prevalent in the MS group (Wilcoxon rank sum test, $P < 0.05$). It is worth noting that the common category is formed by similar microbial reads from the donor and pre-FMT metagenomes, which the sorting algorithm could not clearly

distinguish. This category is prevalent in the MS group in comparison with the healthy and CDI groups (Wilcoxon rank sum test, $P < 0.001$). Interestingly, the control sample set analysis shows the high microbiota stability over time. According to this analysis, the main part of the microbial diversity comes from the baseline sample. At the same time, the almost complete absence of the acquired category in the control group indicates that the sensitivity and specificity of the RECAST make it possible to achieve a high quality of the classification using real metagenomes.

In addition, in our analysis, we highlighted the external read category which consists of microbial genomes whose origin could not be determined. These may be metagenomic reads from the uncovered recipient/donor microbes. Also, these may be reads that should have entered the acquired or survived categories but were misclassified. In addition, it could also be explained by transient microflora. There was a decreased external category in the control group in comparison with real FMT (Wilcoxon rank sum test, $P < 0.05$). It should be noted that in post-FMT samples of the control data set, "donor" microbiota was not found (as expected), whereas the intersection between the "donor" and recipient microbes was ~25%. The ratio of the categories must be relatively stable. Thus, we can conclude that the gut microbiota has a stable unique structure and overlaps up to ~25% between two independent persons.

For additional validation of the observed biological effects, the basic analysis using a set of generally accepted metagenomic approaches and original (nonsorting) metagenomes was performed. Taxonomic annotation was performed using MetaPhlAn2, and microbial strain profiling was performed by metaSNV. The obtained results are presented in Fig. S3. The variability of species/strains over time was significantly higher in the healthy, CDI, and MS data sets than in the control data set (Wilcoxon rank sum test, $P < 0.01$). The distance based on species and strain levels from the donor sample decreased over time in the healthy and CDI sample sets, whereas the MS sample set showed a strong decrease 2 days after transplantation, followed by a gradual increase. Thus, the data obtained indicate a strong effect of the donor microbiota on the recipient's gut microbiota profile after FMT in healthy and CDI sample sets; however, in the MS allogeneic sample set, this effect is reduced.

Thus, results produced using common metagenomic methods such as MetaPhlAn2 or MetaSNV and RECAST are similar. The change in the Bray-Curtis dissimilarity and Manhattan distance to the donor samples corresponds to the increase of the donor fraction in the post-FMT metagenomic samples. However, the RECAST algorithms allow the determination of donor/recipient microbe rate contribution to recipient microbiome assembly after FMT. Moreover, using the RECAST approach, the functional composition of read categories can be studied. These results are presented further.

**(iv) Discovery of functional features associated with restructuring.** To identify potential functional features associated with gut microbiota restructuring due to FMT, functional profiles of read categories were obtained using the HUMAnN2 pipeline. In total, 5,199 Kyoto Encyclopedia of Genes and Genomes (KEGG) orthology (KO) groups in all categories of reads were identified. Functional differences were determined strictly between dependent read categories, between colonizers/noncolonizers or resistant/suppressed separately. Differences in KO profiles between read categories are presented in Fig. 4 and Tables S6 and S7. According to this analysis, the colonizer category is positively associated with 10 KO, whereas the resistant category is positively associated with 25 KO. However, more gene groups were associated with the noncolonizer and suppressed categories.

The top 10 KO by $\log_2$ fold change associated with read categories is shown in Fig. 4B. Interestingly, KO associated with both colonizer and resistant categories had lantibiotic/antibiotic resistance. The overrepresented KOs in both groups included K20492 (lantibiotic transport system permease protein, NisG), K20491 (lantibiotic transport system permease protein, NisE), K06132 (cardiolipin synthase C), K18220 (ribosomal protection tetracycline resistance protein), K20490 (lantibiotic transport system ATP-binding protein, NisF), K19545 (lincosamide nucleotidyltransferase A/C/D/E),

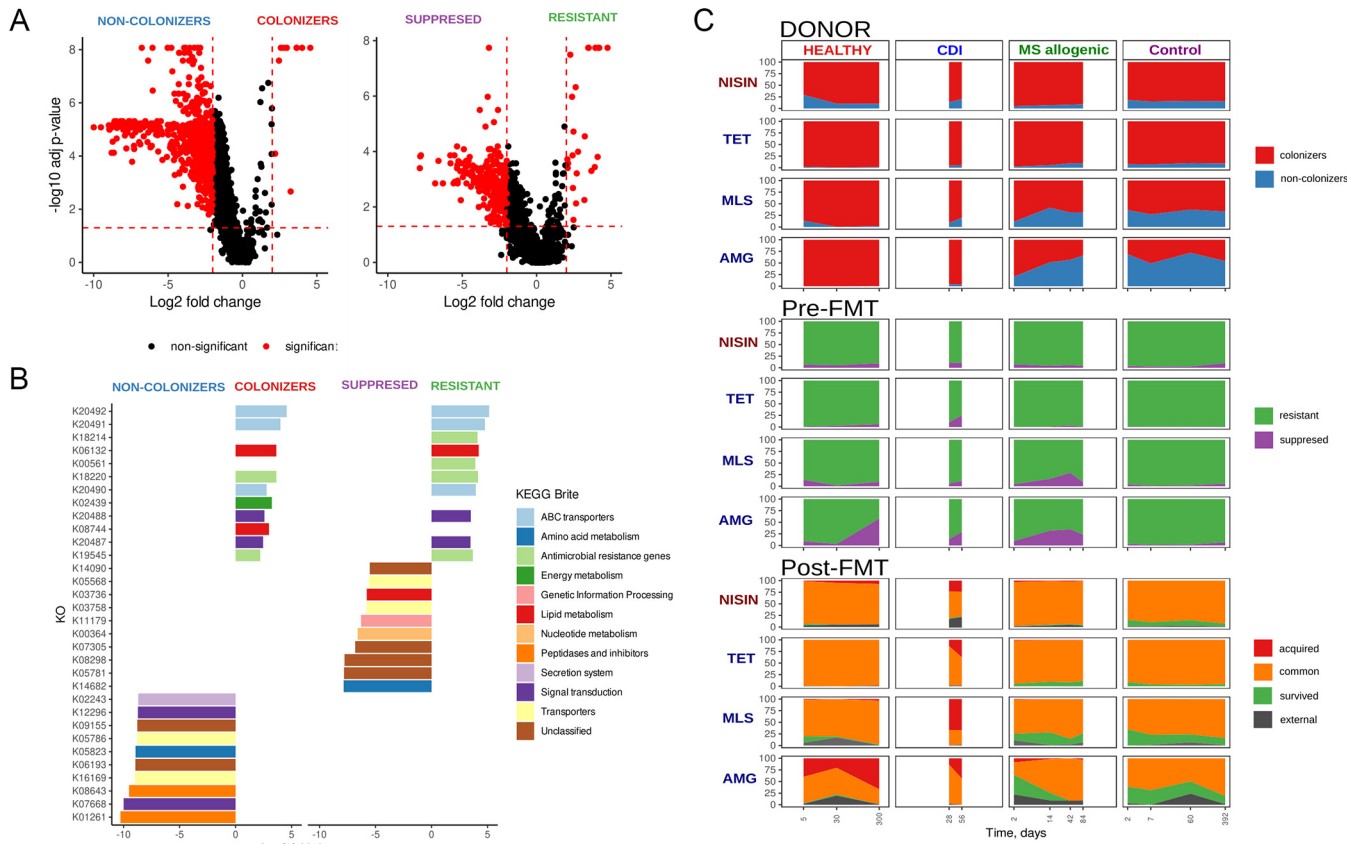

**FIG 4** Functional analysis of gut microbiota restructuring during FMT. (A) Volcano plots showed differences in KEGG orthology (KO) groups) content between two sets of read categories produced by the RECAST algorithm: colonizers and noncolonizers or resistant and suppressed. The $x$ axis denotes $\log_2$ fold change of KO RPK value produced by the HUMAnN2 pipeline. The $y$ axis denotes adjusted $P$ value obtained using Wilcoxon signed-rank test with FDR correction for multiple testing as a result of comparing the KO level between the same read categories. (B) The KO groups differentially distinguish read categories. The KO groups with negative $\log_2$ fold change are associated with noncolonizers and suppressed, while colonizers and resistant KOs have a positive effect size. (C) Distribution of antibiotic and nisin resistance genes in read categories over time. The related metagenomic sample for all categories is indicated above the graphs. The antimicrobial group of genes is indicated to the left of the graphs. The read categories are indicated by different colors. The following abbreviations have been adopted for antibiotic resistance gene groups: TET, tetracyclines; MLS, macrolide, lincosamide and streptogramin; AMG, aminoglycosides.

K20488 (lantibiotic biosynthesis response regulator NisR/SpaR), K20487 (lantibiotic bio-synthesis sensor histidine kinase NisK/SpaK), K08744 (cardiolipin synthase [CMP-form-ing]), and K02439 (thiosulfate sulfur transferase).

Furthermore, extended analysis of antibiotic (using the MEGARes 2.0 antibiotic resist-ance genes [ARGs] database) and nisin (using 5 KOs from the KEGG database described above), resistance gene enrichment in obtained read categories was carried out (Fig. 4C). According to the results obtained, the control sample showed a similar effect: the tetra-cyclines/nisin resistance genes were overrepresented in colonizer and resistant catego-ries compared with the noncolonizer and suppressed sets. Interestingly, in the post-FMT metagenomes, groups of tetracyclines and nisin resistance genes were prevalent in the common read category in both real FMT and control data. Thus, the resistance to tetra-cyclines and nisin can be a common characteristic of the human gut microbes and can be related to forming post-FMT recipient's metagenomes.

## DISCUSSION

Fecal microbiota transplantation (FMT) is currently used to treat recurrent *Clostridioides dif-ficile* infection and other diseases. On the other hand, FMT can be considered a useful model for studying the ecology of host-associated microbial communities. After the FMT procedure, the restructuring processes of the recipient's gut microbial community are observed. How do donor-derived and recipient-derived microbes contribute to the microbiota reassembly after

FMT? Some researchers have tried to answer this question using comparative analysis of gut microbiota taxonomy of donors and recipients obtained by the 16S rRNA gene sequencing approach (13–18). The authors noted the shift of recipients' microbial profiles after FMT toward the donors' profiles. Furthermore, the fact of colonization by donor microorganisms of the recipient's intestines has been established using a combination of stool sample metagenomic sequencing and improved computational approaches (8–12). However, only the behavior of donor strains has been demonstrated. What happens to the recipient's microbial diversity?

Using the proposed RECAST algorithm, the donor-derived and recipient-derived microbes that formed the recipient post-FMT stool metagenomes were identified. According to our analysis, these microbes have a higher relative abundance in the human population metagenomes compared to noncolonizer donor-derived microbes or recipient-derived microbes which are lost by recipients of FMT. It is worth adding that these results were similar between all samples included in analysis and do not depend on disease.

These results are consistent with previous studies. The donor's metagenome-assembled genomes that colonized all recipients were prevalent, and the ones that colonized neither were rare across the participants of the Human Microbiome Project samples in a FMT study of CDI patients (10). Other research states that engraftment can be predicted largely from the abundance and phylogeny of bacteria in the donor and the pre-FMT recipient (11).

Moreover, the functional analysis allows us to determine gene groups that may be associated with FMT-mediated gut microbiome restructure. The donor-derived and recipient-derived microbes which formed the post-FMT recipient's metagenomes were enriched by nisin, tetracycline, lincosamide, and aminoglycoside resistance genes. These observations may be associated with previously described colonization resistance phenomena (19–22). According to this hypothesis, antibiotics can be produced by gut microbiota and form one of the resistance mechanisms against colonization by third-party bacteria. It is worth noting that *Blautia obeum* is a producer of nisin O, which was isolated from human gut microbiota. This research adds to the evidence that lantibiotic production may be an important trait of gut bacteria (23).

Gut microbiota produce a broad spectrum of antimicrobials (24), which can be included in the development of protection mechanisms against colonization by pathobionts and other third-party bacteria (25, 26). Likewise, tetracycline antibiotic resistance genes were found within the Hadza hunter-gatherer population in Tanzania, which was not exposed to anthropogenic pressure in comparison to the residents of modern urban areas (27). This additionally confirms the ecological role of these genes in the human intestine microbiota.

Another possible explanation for the accumulation of lantibiotic resistance genes may be cross-resistance to human antimicrobial peptides (28, 29). In this way, detected characteristics of human gut microbes can be associated with resistance to human-derived antimicrobial peptides (30). On the other hand, lantibiotic/antibiotic resistance gene accumulation in the gut microbiota can be caused by exogenous reasons, including systematic exposure to the foodborne nisin or other antibiotics (31).

On the basis of the results obtained, we assume the existence of a fundamental biological rule mediating donor-derived microbe colonization phenomena. The "restructuring hypothesis" can be formulated: the most prevalent of the donor stool microbiota and likely the most prevalent in the human population stool microbiota can colonize the recipient's intestine. This hypothesis can be extended: the recipient tends to retain its prevalent gut microbiota. The bold assumption may be that the most prevalent microbes form the "core" of the human gut microbiota which is relatively stable over time. It seems that the donor "core" stool microbiota modifies the "core" of the recipient gut microbiota due to FMT. "Core" microbe resistance to microbe-derived and potentially human-derived antimicrobials mediates this process. Post-FMT microbiota are formed by donor and recipient microbial "core" competitions. The more competent

mSystems®

donor/recipient microbes will gain an advantage over the recipient/donor bacteria, which will affect the post-FMT metagenome profile (32, 33).

On the other hand, the proposed "restructuring hypothesis" may be a consequence of the fact that by using sequencing of total stool DNA, we observe not only the gut microbiota but also transient bacteria (from food, drinking water, or other environmental sources) that are not gut microbiota residents. These transient bacteria may be a significant part of gut microbiota diversity. Perhaps, the entire "true" intestinal microbiota of the donor can colonize the recipient's intestines with various degrees of success. On the other hand, biological sample collection/preparation, sequencing artifacts and/or taxonomic classification and/or other reasons may affect the quality of the analysis and hence the reliability of conclusions. In any case, these observations certainly require additional confirmation.

**Conclusions.** Here, we presented a novel computational approach RECAST to track the restructuring process of the gut microbiota due to FMT. The method is based on sorting post-FMT recipient's metagenomic reads by origin from the donor or recipient microbiome. The functional analysis of the obtained read categories revealed the enrichment of successful gut microbes by lantibiotic/antibiotic resistance genes. The results obtained with publicly available data sets allowed us to propose the "restructuring hypothesis": the most prevalent of the donor stool microbiota and likely the most prevalent in the human population stool microbiota can colonize the recipient's intestine. To summarize, this approach allows researchers to gain novel biological insights via providing the improved resolution of FMT study analysis.

## MATERIALS AND METHODS

**Read classification algorithm.** We developed the RECAST (recipient intestine colonization analysis tool) algorithm, based on MetaCherchant source code (34), to compare two metagenomes and extract reads of one metagenome found in another. It takes as input two samples with paired-end reads in fasta or fastq format. One of the samples is referred to as queried, the other as analyzed. In the first stage, the program retrieves all $k$-mers from the queried metagenome and saves the quantity of each $k$-mer in a data structure referred to later in this article as the de Bruijn graph. In the second stage, each pair of reads from the analyzed metagenome is examined, and both reads are searched independently in the queried metagenome. All $k$-mers (substrings of length $k$) are extracted from the read and are searched for in the de Bruijn graph. As a result, mean depth coverage of a read by $k$-mers as well as breadth coverage of a read is obtained. Breadth coverage is defined as a proportion of positions in read, covered by $k$-mers from de Bruijn graph. Then theoretical estimation of breadth is used to classify each read as found or not found in the de Bruijn graph.

Given the mean depth coverage, the theoretical breadth coverage estimation is required for comparison to the calculated one. Onwards we are following (35). Let us assume that the number of $k$-mers covering a fixed position in the read obeys a Poisson distribution with probability mass function

$$p(n) = \frac{\lambda^n}{n!} e^{-\lambda}$$

where $\lambda$ is the mean depth coverage. This assumption is reasonable because the read is covered evenly, and there are no jumps in coverage. Hence, the probability of a position in the read being not covered is

$$p(0) = e^{-\text{meanCoverage}}$$

Consequently, theoretical breadth can be found as

$$\text{theoryBreadth} = 1 - e^{-\text{meanCoverage}}.$$

Having calculated the theoretical breadth, we next defined the confidence interval, which contains the reads classified as found. It was approximated using the central limit theorem. Breadth coverage is in the range

$$(\text{theoryBreadth} - \delta; \ \text{theoryBreadth} + \delta)$$

at 95% confidence level for:

$$\delta = 1.96 \ \frac{\sqrt{p(0)(1 - p(0))}}{\sqrt{\text{length}}}$$

To further control the quality of found reads, we introduced a threshold for minimal breadth coverage (0.9 by default). Summing up, the read is classified as found if it both satisfies the threshold and falls

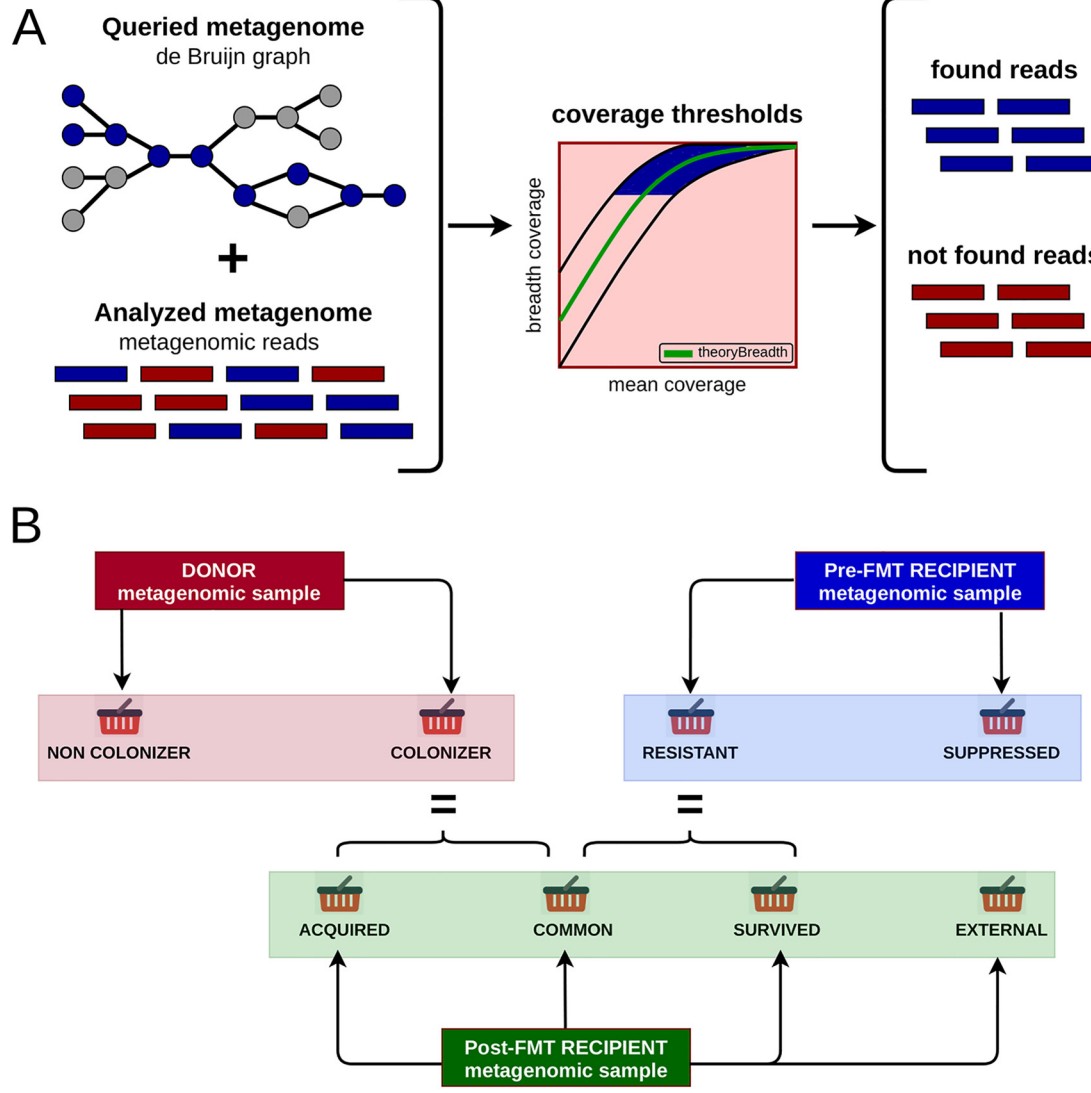

**FIG 5** The reads classification algorithm workflow. (A) The basic scheme of the reads classification algorithm. (B) The scheme of obtaining metagenomic read categories.

within the confidence interval. Otherwise, the read is classified as not found. During the processing of paired-end reads, reads from some pairs can be classified into different categories. This indicates the discrepancy between the classification and paired nature that might have been caused by small genome variations or sequencing errors. These read pairs are not credible and are excluded from further analysis. The schematic workflow of the algorithm is presented in Fig. 5A.

**FMT read classification by origin.** A design of FMT experiments to study the behavior of the gut microbiota usually involves the collection and sequencing of stool samples of the donor, the recipient before FMT (pre-FMT sample), and the recipient after FMT (post-FMT sample). The RECAST algorithm takes as input every two out of three metagenomic samples described above and splits reads from each metagenome into different categories based on the origin in the recipient's post-FMT metagenome. First, donor reads are queried against a post-FMT recipient sample to generate colonizer and noncolonizer categories. Second, pre-FMT recipient reads are queried against the post-FMT recipient sample to generate categories resistant to FMT and suppressed by FMT. Third, post-FMT recipient reads are queried against the donor sample and split into two temporary categories: found and not found. Further, these categories are queried against the pre-FMT sample to split post-FMT reads into four categories: acquired via FMT (reads found only in donor sample), common (reads found in both donor and pre-FMT samples), survived FMT (reads found only in the pre-FMT sample), and external (reads not found in donor or pre-FMT samples). A diagram of the produced read categories is presented in Fig. 5B. The pseudocode of the algorithm is shown in Tables 1 and 2.

**Simulated data.** The RECAST algorithm validation using simulated data was performed in two steps. First, the set of *Escherichia coli* strain genomes with different nucleotide distances was used for simple

**TABLE 1** Algorithm 1: RECAST algorithm workflow

| RECAST algorithm workflow |
| --- |
| 1: (**colonizers, noncolonizers**) = ReadsClassifier(queried=Post-FMT, analyzed=Donor) |
| 2: (**resistant, suppressed**) = ReadsClassifier(queried=Post-FMT, analyzed=Pre-FMT) |
| 3: (found, nonfound) = ReadsClassifier(queried=Donor, analyzed=Post-FMT) |
| 4: (**common, acquired**) = ReadsClassifier(queried=Pre-FMT, analyzed=found) |
| 5: (**survived, external**) = ReadsClassifier(queried=Pre-FMT, analyzed=nonfound) |

reads simulations by the InSilicoSeq tool (36) with standard Illumina HiSeq error pattern. The variation in strains was assessed by nucleotide proximity using Mash distance (37). According to the Mash paper (37), average nucleotide identity (ANI) ≈ 1 − Mash distance. We used the 1 − Mash distance score for assessing nucleotide similarity of *E. coli* genomes. The most probable scenario of strains behavior due to FMT was modeled: donor and recipient strains coexist in the recipient's intestine (8). The *E. coli* strains used are presented in Table S1 in the supplemental material.

At the second step of simulations, we used 1,520 reference genomes from cultivated human gut bacteria (38). The artificial metagenomes were simulated with different complexity (30, 100, or 300 genomes per metagenome were selected using random subsampling) and mean read coverage (5× to 80×). Post-FMT recipient metagenomes were formed by the mixture of "donor" and "recipient" genomes, as well as "external" genomes (nondonor and nonrecipient genomes) for increased classification complexity. The post-FMT artificial recipient's metagenome included 40.5% ± 0.5% donor-derived unique genomes, 36.1% ± 5.9% recipient-derived genomes, 8.0% ± 8.3% common genomes, and 15.5% ± 10.7% external genomes (Table S1). To further complicate the classification, we added a few identical genomes in both the "donor" and the "recipient" subsets. Thus, these genomes should be classified as "common." The simulation was performed using the InSilicoSeq tool with a standard Illumina HiSeq error pattern. The bacterial genomes used are presented in Table S1.

**Real metagenomic data upload and quality control.** The experimental FMT data used in this study are longitudinally collected recipient metagenomes (one time point before transplantation and several after), as well as associated donor metagenomes. All whole-genome sequencing (WGS) metagenomes containing both donor and recipient samples available at the start of the study were selected. An additional criterion for data selection was the presence of aligned sampling points between recipients.

The metagenomic data from FMT-allogeneic experiments in healthy volunteers (12) (healthy group), patients with *Clostridioides difficile* infection (10) (CDI group) and metabolic syndrome (39) (MS group) were examined in the study. Additionally, metagenomic data from healthy people without interventions

**TABLE 2** Algorithm 2: ReadsClassifier main routine

| Step/line no. | ReadsClassifier main routine |
| --- | --- |
| | input: two sets of metagenomic reads: *queried* metagenome to search in and *analyzed* metagenome to classify its reads by origin. |
| | output: two sets of metagenomic reads which are subsets of analyzed metagenome: those found in the queried metagenome and those not found in the queried metagenome. |
| 1 | Initialize data structures: |
| 2 | both_found – queue storing pair of reads, both of which are found in queried metagenome |
| 3 | first_found – queue storing pair of reads, first of which are found in queried metagenome |
| 4 | second_found – queue storing pair of reads, second of which are found in queried metagenome |
| 5 | none_found – queue storing pair of reads, none of which are found in queried metagenome |
| 6 | Read metagenomic data from files specified in queried parameter |
| 7 | Store all *k*-mers in a hash map: *k*-mer → its coverage |
| 8 | Create a *thread pool* |
| 9 | for each *pair of reads* (R1, R2) *from analyzed metagenome using thread pool* do |
| 10 | if *R1 satisfies coverage threshold and R2 satisfies coverage threshold* then |
| 11 | add (R1, R2) to both_found |
| 12 | else if *R1 satisfies coverage threshold and R2 does not satisfies coverage threshold* then |
| 13 | add (R1, R2) to first_found |
| 14 | **else if** *R1 does* **not** *satisfies coverage threshold* **and** *R2 satisfies coverage threshold* **then** |
| 15 | add (R1, R2) to second_found |
| 16 | **else** |
| 17 | add (R1, R2) to none_found |
| 18 | **end** |
| 19 | **end** |
| 20 | Save all analyzed *k*-mers in files with respect to their classification |
| 21 | **return** (both_found, none_found) *k*-mers |

**TABLE 3** Metagenomic data sets used in the study

| Data set | No. of all individuals/ no. of samples | No. of donors/ no. of samples | No. of recipients/ no. of samples | No. of reads per metagenome (mean ± SD), mln | Sequencing platform (read length [bp]) |
|---|---|---|---|---|---|
| Healthy | 4/17 | 1/3 | 3/14 | 23.3 ± 3.7 | Illumina (250) |
| CDI | 3/10 | 1/4 | 2/6 | 57.2 ± 13.4 | Illumina (150) |
| MS | 8/30 | 3/5 | 5/25 | 56.6 ± 18.0 | Illumina (100) |
| Control | 5/27 | | | 73.1 ± 38.8 | Illumina (100) |
| HMP 2012 | 139/139 | | | 105.1 ± 19.9 | Illumina (100) |

were used (40) (control group) as a benchmark group. Including the control group will allow us to adequately distinguish between effects associated with FMT and effects associated with natural variations of metagenomic data. For additional computational experiments, 139 metagenomic stool samples from the HMP 2012 data set (41) were used. In total, 223 real metagenomic stool samples were used in the study. Description of the data sets and basic statistics are presented in Table 3 and Table S2.

Raw metagenomic data were downloaded from public repositories using fastq-dump from the SRA Toolkit (42), quality assessment was performed with FastQC (https://github.com/s-andrews/FastQC). Technical sequences and low-quality bases were trimmed with the Trimmomatic tool (43). The threshold for sequencing quality was set to $Q > 30$. The human sequences from metagenomic samples were removed by bbmap (44) using GRCh37 human genome version (https://www.ncbi.nlm.nih.gov/assembly/GCF\_000001405.13). Described metagenomics read preprocessing computational steps were implemented in the Assnake metagenomics pipeline (https://github.com/ASSNAKE). The preprocessing results are presented in Table S2.

After quality control, samples were sorted using the RECAST algorithm by the categories described above. As a control, sorting was also performed in the control group. Each baseline metagenome was selected as a "donor sample," while the remaining metagenomes from this subject were not used in the analysis. In total, the 10 sorting series were performed in real FMT data sets and 20 sorting series were used in the control data set. Each sorting series consists of several algorithm runs—one for each post-FMT time point. In total, there were 105 sorting procedures: 33 for real FMT data, 72 for control data.

**Data analysis and visualization.** After processing the real metagenomes using the RECAST algorithm, the read categories were characterized using common metagenomic computational approaches. Taxonomic profiles were obtained by the MetaPhlAn2 tool (45, 46). Additional visualizations were performed using vegan package (47) with Bray-Curtis dissimilarity and metaMDS function with default parameters and with the ggplot2 library (https://ggplot2.tidyverse.org) implemented in GNU/R. PERMANOVA (adonis function from the vegan package for GNU/R) and Bray-Curtis dissimilarity (47) tests were used as measures for comparing taxonomy profiles of read categories.

Functional profiles were obtained via the HUMAnN2 pipeline (48) and the KEGG database (release 2018-03-26) (49). Log$_2$ fold change in KEGG orthology (KO) group levels were calculated. The absolute value (modulus) of the log$_2$ fold change threshold was set at 2. Wilcoxon signed-rank test with false discovery rate (FDR) correction for multiple hypothesis testing was used to determine significant differences in functional profiles (adjusted $P < 0.05$). Antibiotic resistance genes (ARGs) were identified in the metagenomes by mapping the metagenome reads to MEGARes 2.0 database (50) using Bowtie2 (51). Read counts of ARGs were calculated using the ResistomeAnalyzer tool (52).

The Random Forest algorithm was used to order microbial feature contribution to the gut microbiota restructuring process. Metadata (recipient subject, donor subject, sampling time, and disease) or taxonomic (phylum, class, order, family) features, and the relative abundance of microbes in the donor or recipient metagenomic samples as well as in the HMP 2012 data set were added in this analysis. The classification of the pairs of read categories such as colonizers/noncolonizers and resistant/suppressed was performed separately because they derive from different samples: donor and pre-FMT recipient metagenomes, respectively. Microbial species whose MetaPhlAn2 markers were covered with fewer than 100 reads were not included in the analysis. For model building, parameter optimization, and associated data processing, pandas, numpy, and scikit-learn libraries for python 3 and jupyter-lab were used.

Nonsorted metagenomic reads of experimental data sets were examined using common metagenomic approaches such as MetaPhlAn2 and metaSNV (53) profiling based on mOTUs2 database (54). Bray-Curtis dissimilarity and Manhattan distances were used as measures for comparing taxonomic and strain profiles of nonsorted metagenomes, while a Wilcoxon rank sum test was applied for identifying significant differences in these profiles.

**Availability of data and materials.** The study used data from open sources, which are available at NCBI Sequence Read Archive under the BioProject accession numbers PRJNA510036, PRJEB12357, and PRJNA353655, at European Nucleotide Archive (ENA) database ERP009422, and at https://www.hmpdacc.org. The source code for the RECAST algorithm can be found at https://github.com/ivartb/RECAST.

## SUPPLEMENTAL MATERIAL

Supplemental material is available online only.

**FIG S1**, PDF file, 0.4 MB.

**FIG S2**, PDF file, 0.2 MB.
**FIG S3**, PDF file, 0.2 MB.
**TABLE S1**, XLSX file, 0.01 MB.
**TABLE S2**, XLSX file, 0.01 MB.
**TABLE S3**, XLSX file, 0.2 MB.
**TABLE S4**, XLSX file, 0.5 MB.
**TABLE S5**, XLSX file, 0.02 MB.
**TABLE S6**, XLSX file, 0.03 MB.
**TABLE S7**, XLSX file, 0.02 MB.

## ACKNOWLEDGMENTS

We thank Alexander I. Manolov for useful comments, Dmitry E. Fedorov for help in metagenomic data preprocessing and taxonomic annotation, and Andrey E. Samoilov for help in the statistical justification of the sorting algorithm.

E.I.O. and E.N.I. were financed by the funds of the state assignment "Assessment of the contribution of chronic maternal diseases to the dynamics of changes in the biodiversity of the intestinal microbiota of premature infants in the first weeks of life" (NID: INFANTS). A.B.I. and V.I.U. were supported by JetBrains Research. Also, we thank the Center for Precision Genome Editing and Genetic Technologies for Biomedicine, Federal Research and Clinical Center of Physical-Chemical Medicine of the Federal Medical Biological Agency for providing computational resources for this project.

E.I.O. performed computational experiments using simulated and real metagenomic data, interpretation of obtained results, performed data analysis and visualization, contributed to algorithm design, contributed to research idea, and wrote the manuscript. A.B.I. and V.I.U. designed and implemented the RECAST algorithm and contributed to manuscript preparation. A.B.I. performed computational experiments using simulated data, partially wrote the manuscript, and contributed to interpretation of the obtained results and data analysis. E.N.I. came up with the research idea and contributed to interpretation of the obtained results and manuscript preparation.

We declare that we have no competing interests.

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
