## [Reviewer comments · mSystems]

Separation of donor and recipient microbial diversity allows determining taxonomic and functional features of gut microbiota restructuring following fecal transplantation

Evgenii Olekhnovich, Artem Ivanov, Vladimir Ulyantsev, and Elena Ilna

Corresponding Author(s): Evgenii Olekhnovich, Federal Research and Clinical Centre of Physical and Chemical Medicine

Review Timeline:

Submission Date:

June 24, 2021

Accepted:

July 18, 2021

Editor: Pedro Oliveira

Reviewer(s): Disclosure of reviewer identity is with reference to reviewer comments included in decision letter(s). The following individuals involved in review of your submission have agreed to reveal their identity: Junjun Wang (Reviewer #2)

Transaction Report:

DOI: <https://doi.org/10.1128/mSystems.00811-21>

July 18, 2021

Dr. Evgenii I. Olekhovich
Federal Research and Clinical Centre of Physical and Chemical Medicine
Moscow
Russia

Re: mSystems00811-21 (Separation of donor and recipient microbial diversity allows determining taxonomic and functional features of gut microbiota restructuring following fecal transplantation)

Dear Dr. Evgenii I. Olekhovich:

Your manuscript has been accepted, and I am forwarding it to the ASM Journals Department for publication. For your reference, ASM Journals' address is given below. Before it can be scheduled for publication, your manuscript will be checked by the mSystems senior production editor, Ellie Ghatineh, to make sure that all elements meet the technical requirements for publication. She will contact you if anything needs to be revised before copyediting and production can begin. Otherwise, you will be notified when your proofs are ready to be viewed.

As an open-access publication, mSystems receives no financial support from paid subscriptions and depends on authors' prompt payment of publication fees as soon as their articles are accepted. =

Publication Fees:

We recognize that the video files can become quite large, and so to avoid quality loss ASM suggests sending the video file via <https://www.wetransfer.com/>. When you have a final version of

the video and the still ready to share, please send it to Ellie Ghatineh at eghatineh@asmusa.org.

Sincerely,

Pedro H. Oliveira
Editor, mSystems

Journals Department
Table S4: Accept
Supplemental Material: Accept
Table S7: Accept
Table S3: Accept
Table S6: Accept
Supplemental Material: Accept
Table S1: Accept
Table S2: Accept
Table S5: Accept
Supplemental Material: Accept